# Adversarially Robust Latent Bandits in Multiplayer Asymmetric Settings

## Abstract

We examine a novel multiplayer extension of the latent multi-armed bandit problem as formulated in Maillard & Mannor (2014), with broad applications such as recommendation systems and cognitive radio. Following Chang et al. (2022), we examine three information asymmetric scenarios: Problem A, in which players receive identical rewards but cannot observe each other's actions; Problem B, players receive private i.i.d rewards but can observe others' actions; and Problem C, players receive private i.i.d rewards and cannot observe others' actions. For problems A and B, we provide nearly optimal gap-independent regret bounds. When reduced to the single agent setting, our results improve on Maillard & Mannor (2014) by allowing for adversarial nature's actions. For Problem C, we use the knowledge of the reward means to improve on the results in Chang et al. (2022).

## 1 Introduction

The single-player multi-armed bandit problem is a general reinforcement learning problem where a single player has access to multiple arms. Each arm is associated with an unknown reward distribution, varying potentially in rewards, and the agent attempts to maximize cumulative reward by observing the empirical rewards, which converge to the true mean of each arm. Maillard & Mannor (2014) proposes the latent multi-armed bandit problem. This has applications in recommendation systems (Li et al. (2010)), where the agent selects an advertisement to show each arriving user, using a context vector that captures information such as the user's browsing behavior or geographic location. In cognitive radio (Avner et al. (2012)), the agent must choose a communication channel based on its current location and network status, while avoiding interference with other sources like radar or WiFi. However, relying solely on observable context may not be enough to achieve optimal solutions. In recommendation systems, key details like a user's gender or income are often unavailable due to privacy concerns. Similarly, in cognitive radio settings, it is unclear whether other sources or users are nearby or distant. In both scenarios, crucial aspects of the reward structure remain unobserved, but the latent structures can be inferred. In this paper, we extend latent bandits to the multiplayer setting.

Multi-agent reinforcement learning (MARL) has been applied to various domains such as autonomous driving (Shalev-Shwartz et al. (2016)), strategic board games like Go (Silver et al. (2016)), real-time strategy games, robotic control (Kober et al. (2013)), and card games (Brown et al. (2017); Brown & Sandholm (2019)). These successes are often enabled by deep neural networks and frequently involve multiple agents, highlighting the importance of studying MARL — where autonomous agents interact within a shared environment to maximize long-term rewards (Busoniu et al. (2008)). Beyond gaming, MARL is increasingly relevant in domains like cyber-physical systems (Wang et al. (2016)), finance (Lee et al. (2007)), sensor networks (Choi et al. (2009)), and social sciences (Leibo et al. (2017)).

The main drawback in previous multi-agent works is that they usually allow for communication or do not consider joint actions. In the case of games where both of these assumptions are not necessary, they do not seek to find a global optimal action for the players but rather some type of equilibrium. Therefore we consider the line of work that addresses partial or no observability of other players' actions, rewards or both. To address the complexity of multi-agent systems, Chang et al. (2022); Chang & Lu (2023) proposes algorithms

where players face information asymmetry in rewards, actions or both; Chang & Lu (2025) and Chang & Karthik (2025) extend this framework to contextual and metric bandits, achieving sub-linear upper bounds on regret. We address the limited-communication setting by leveraging players' ability to coordinate before online learning begins; we also account for the fact that one player's actions may influence another's rewards through the analysis of *joint actions*.

We also examine three cases of information asymmetry, in which players do not observe others' rewards or actions but must coordinate to minimize their regret without utilizing an explicit communication channel. We investigate three types of information asymmetry: In Problem A, we are faced with the problem of asymmetry in actions, where the players cannot see each other's actions but receive identical rewards. Problem B is the opposite, as players can see each other's actions but receive private i.i.d rewards. Finally, in Problem C, players are unable to see others' actions and receive private i.i.d rewards.

**Our contribution**   First, we utilize the framework proposed in Chang et al. (2022) to extend the multiple-cluster arrival problem in Maillard & Mannor (2014) to a novel multi-agent setting, accounting for limited communication and information asymmetry between agents. Second, we show that our results improve on the work of Maillard & Mannor (2014) in the single-agent case by allowing for adversarial, rather than stochastic, nature's actions. Finally, we provide a gap-independent regret bound which also holds for the single-agent case, improving on Maillard & Mannor (2014). In section 3.1, we show that the cumulative regret of `Multiple-K-Intervals-A` under problem A satisfies $\mathcal{R}_T \leq O(\sqrt{T \log T} BC)$. In section 3.2, we propose `Multiple-K-Intervals-B` for problem B, which also satisfies $\mathcal{R}_T \leq O(\sqrt{T \log T} BC)$. In section 3.3, we propose `Multiple-K-mDSEE` for problem C, with $\mathcal{R}_T \leq O(BC \log(T)/(\min_{a^\star,b} \Delta_{a^\star,b})^2)$.

**Related Works   Latent Bandits**: Agrawal et al. (1989) provides an early formulation of the multi-armed bandit with latent cluster (MAB-LC) problem in which reward distributions are parametrized by an unknown parameter from a known parameter space, providing a lower bound significantly different from the standard bound for multi-armed bandits.

We focus on latent bandits as introduced in Maillard & Mannor (2014), where nature's actions belong to clusters with known reward distributions (both spaces being discrete and finite). Maillard & Mannor (2014) considers the setting in which all states belong to a single cluster and propose `Single-K-UCB`, which improves on Agrawal et al. (1989). The authors also consider a setting in which nature's actions can belong to any of several known clusters, achieving gap-dependent sublinear regret with `Multiple-K-UCB`; this is the problem we examine below.

Many works explore regret minimization for latent bandits (Pal et al. (2023b;a); Hong et al. (2020a)). Kinyanjui et al. (2023) addresses fixed-confidence pure-exploration for latent bandits. Another line of work (Hong et al. (2020b); Gentile et al. (2014); Zhou & Brunskill (2016); Bresler et al. (2014)) applies the latent bandit problem to recommendation systems, using a combination of offline learning to infer latent structures and online learning to provide personalized recommendations.

Kausik et al. (2024) and Gupta et al. (2020) investigate more structured variants of the MAB-LC problem; Gupta et al. (2020) assumes rewards are known functions of a common latent variable. However, Gupta et al. (2020) assumes that the latent random variable follows a fixed distribution; in contrast, we make no assumptions about the distribution of the latent variable. Similarly, Kausik et al. (2024) assumes a fixed distribution for the latent variable and a linear contextual framework. By loosening these assumptions, we allow for more adaptivity to changes in the latent variable. Further, in contrast with Kausik et al. (2024) and other contextual bandit works, we do not assume that there is a fixed function class relating rewards to the context-action feature map.

**Multiplayer Bandits:** In cooperative multiplayer bandit problems, players collectively identify the best arm from a shared set. Communication between players is often modeled by a graph structure (Awerbuch & Kleinberg (2008)). Various strategies have been developed, including $\epsilon$-greedy methods (Szorenyi et al. (2013)), gossip-based UCB variants (Landgren et al. (2016); Martínez-Rubio et al. (2019)), and leader-based coordination (Wang et al. (2020)). Other studies such as Cesa-Bianchi et al. (2016) allow players to send and receive messages with some delay. In the above papers, explicit communication (whether between all players

or a subset) is assumed, whereas we consider limited communication scenarios where players cannot see each other's rewards, actions, or both. Rather than explicit messages between players, we utilize players' ability to coordinate before online learning begins. Cesa-Bianchi et al. (2020) considers environments where only certain players are active in each round and players act asynchronously; in contrast, we require that each player takes an action at each round.

Motivated by cognitive radio, where the environment may fluctuate unpredictably, there is a line of work on adversarial multiplayer bandits (Howard et al. (2022); Shi & Shen (2021); Bishop et al. (2020); Alatur et al. (2020); Bar-On & Mansour (2019)). These works assume that the arms themselves have adversarial losses; we assume that the rewards are stochastically distributed given an adversarially chosen *nature's* action. Our work assumes that the environment may change rapidly, but each arm's reward is more predictable given that the environment's state can be accurately estimated.

Some works suppose that players do not receive rewards if they select the same arm as another player — a "collision" — which serves as a form of implicit communication similar to our use of "sabotage" in section 3.2 (Proutiere & Wang (2019)). The competing bandits model (Liu et al. (2020)) builds on collision settings by introducing player preferences and hierarchy. In this problem, players report their UCB indices to a central authority; our work does not require any centralized communication. Later work Cen & Shah (2022) showed that optimal logarithmic regret is attainable; Jagadeesan et al. (2021) and Liu et al. (2020) apply this to two-sided markets, and Liu et al. (2021) proposes a decentralized UCB method with built-in collision avoidance. However, the above works assume that players' rewards are independent of the actions of other players in the absence of collisions. In contrast, we allow for joint actions where all players may affect the reward received by others at each round and are incentivized to cooperate. This allows us to consider real-life scenarios where the outcome of an action depends on the choices of surrounding actors, even in the absence of collision.

Competitive MARL, often modeled as zero-sum Markov games (Littman (1994)), captures adversarial dynamics where one agent's gain is another's loss, supporting robust policy development under uncertainty (Zhang et al. (2020)). General-sum games lie between these extremes, featuring agents with differing or conflicting objectives (Basar & Olsder (1999); Bistritz & Leshem (2018)). We instead assume that players have the shared goal of maximizing their collective cumulative reward.

## 2 Preliminary

### 2.1 Single-player latent bandits

First, we present the single-player latent bandit problem, following some notation from Maillard & Mannor (2014). For each round $t$ from 1 to the horizon $T$, nature selects an action $b$ from the set of all possible states $\mathcal{B}$, where $|\mathcal{B}| = B$, which is visible to the player. The player then takes an action $a$ from $\mathscr{A}$ (with $|\mathscr{A}| = K$) and receives a real-valued reward sampled from the distribution $\nu_{a,b}$. We assume that each $\nu_{a,b}$ is 1-sub-Gaussian and $\mathbb{E}[\nu_{a,b}] = \mu_{a,b}$. The set $\mathcal{B}$ is further partitioned into $C$ clusters $\{\mathcal{B}_c\}_{c=1,\ldots,C}$, where the reward distributions $\nu_{a,b}$ for a given action $a$ are identical for all $b$ in a cluster $\mathcal{B}_c$.

### 2.2 Multiplayer latent bandits

We now extend this to the multiplayer problem as examined in Chang et al. (2022), Chang & Lu (2023). Suppose there are $M$ players. The players take a *joint action* $\boldsymbol{a}$, where we express $\boldsymbol{a}$ as a tuple of individual player actions at time $t$, $\boldsymbol{a}_t = (a_1, a_2, \ldots, a_M)$. Each player then receives a real-valued reward sampled from $\nu_{\boldsymbol{a},b}$.

We examine the scenario from Maillard & Mannor (2014) where $\{\nu_{\boldsymbol{a},c}\}$ is known to the players, i.e. $b$ can come from any of several clusters with known reward distributions. Denote $\star_b = \arg\max_{\boldsymbol{a}} \mu_{\boldsymbol{a},b}$, the optimal joint action for each state $b$. Players are not given the true cluster of each state $b$; however, they are given $\star_c = \arg\max_{\boldsymbol{a}} \mu_{\boldsymbol{a},c}$, the optimal arm within each cluster. Thus the problem reduces to identifying the true cluster each state belongs to and we need only explore $\mathscr{A}^\star = \{\boldsymbol{a}^\star \mid \exists c \ \boldsymbol{a}^\star = \star_c\}$, the set of potentially optimal

arms. Maximizing reward is equivalent to minimizing the regret, which we define as

$$\mathcal{R}_T = \sum_{t=1}^{T} \left( \mathbb{E}_{X_t \sim \nu_{\star_{b_t}, b_t}} [X_{\star_{b_t}, b_t}] - \mathbb{E}_{X_t \sim \nu_{a_t, b_t}} [X_{a_t, b_t}] \right)$$

or equivalently, $\mathcal{R}_T = \sum_{t=1}^{T} \left( \mu_{\star_{c_{b_t}}, c_{b_t}} - \mu_{a_t, c_{b_t}} \right)$, where $c_{b_t}$ is the cluster $b_t$ belongs to.

Additionally, denote the number of observations for the pair $(a, b)$ at time $t$ by $N_{a,b}(t) = \sum_{n=1}^{t} \mathbb{I}\{a_n = a, b_n = b\}$ and let the empirical mean built from the same observations be $\hat{\mu}_{a,b}(t)$. Define $N_b(t) = \sum_{a \in \mathscr{A}} N_{a,b}(t)$.

To estimate a confidence interval for each $\mu_{a,b}$ and identify the corresponding cluster for $b$, we construct a confidence interval based on $\hat{\mu}_{a,b}(t)$ (following Maillard & Mannor (2014)). Let $S_{a,b}(t) = (U_{a,b}(t), L_{a,b}(t))$, where

$$U_{a,b}(t) = \hat{\mu}_{a,b}(t) + \sqrt{4 \log(T)/N_{a,b}(t)}$$

and

$$L_{a,b}(t) = \hat{\mu}_{a,b}(t) - \sqrt{4 \log(T)/N_{a,b}(t)}.$$

Note that $\mu_{a,b} \in S_{a,b}(t)$ is true with high probability (see Lemma A). Additionally, if $\mu_{a,c} \in S_{a,b}(t)$ for all $a$ then we consider $c$ to be an admissible cluster for nature's action $b$.

We assume players cannot explicitly communicate with each other during the learning period, but can coordinate on a strategy beforehand. During this strategy meeting, they know the number of actions each player has access to, as well as the horizon $T$. We seek to minimize the cumulative regret across $T$ rounds.

We consider several information asymmetric scenarios as in Chang et al. (2022); Chang & Lu (2023), as follows:

**Problem A: Information Asymmetry in Actions.** In Problem A, we assume that all players receive identical rewards but cannot explicitly observe the actions of other players. We propose a round-robin scheme that allows players to inductively infer each other's actions by maintaining identical admissible sets, enabling consistent updates and avoiding suboptimal joint actions.

**Problem B: Information Asymmetry in Rewards.** In Problem B, we assume that players receive independent and identically distributed (i.i.d.) rewards that are unknown to other players, but can observe the actions of others. We build on the algorithm presented for Problem A with the addition of *sabotage*, which allows players to implicitly signal belief changes regarding the admissibility of clusters.

**Problem C: Asymmetry in Rewards and Actions.** In Problem C, we consider the most information-constrained setting: players receive independent and identically distributed rewards, and observe neither the rewards nor the actions of other players. We propose an explore-then-commit-style scheme run in parallel for each nature's action $b$, where players follow a synchronized exploration schedule based on prior coordination and then independently commit to their estimated optimal arm once sufficient confidence is achieved.

## 3 Main Results

### 3.1 Problem A: Information Asymmetry in Actions

In Problem A, players receive identical rewards but cannot observe others' actions. Therefore, players may become miscoordinated in case of ties between arms. Consider a scenario where there are two players and two optimal actions. Player 1 intends to take $(1, 2)$ whereas Player 2 intends to take $(2, 1)$; since each player controls only their respective entry in the tuple, the resulting joint action taken is $(1, 1)$. Thus, players must agree on an ordering scheme for arms and clusters before the start of learning to ensure that ties are

broken consistently across all players. Any order for arms suffices if all players agree — one option is the lexicographical ordering proposed in Chang et al. (2021). Similarly, players should agree on an ordering for clusters.

As rewards from chosen actions are observed and empirical reward means change, the ranking of arms may change. However, the original ordering remains a constant reference point for resolving any new ties that occur.

**Intuition:** We propose Algorithm 1, which enables players to gradually eliminate non-admissible clusters utilizing confidence intervals $S_{\boldsymbol{a},b}(t)$. Note that we take an optimistic approach, where at round $t = 1$, all clusters are assumed to be admissible for all states. When multiple clusters are admissible for a nature's action, players "test" clusters according to a round-robin scheme as follows, where clusters are ordered as above:

$$c_1 \to c_2 \to \cdots \to c_C \tag{1}$$

The confidence intervals will converge around the true means $\mu_{\boldsymbol{a},b}$, allowing players to identify the true cluster for each nature's action. Because players receive identical rewards and maintain identical confidence intervals, they also maintain identical admissible sets at all times. Thus players remain synchronized and can infer other players' actions, allowing them to update confidence intervals correctly.

We assume that the true mean $\mu_{\boldsymbol{a},b}$ is inside all players' confidence intervals $S_{a,b}(t)$ with high probability, enabling a gap-independent regret analysis based on confidence interval width. Further, Algorithm 1 serves as a baseline that can be modified for multiple types of information asymmetry, as in section 3.2.

---

**Algorithm 1** The `Multiple-K-Intervals-A` Algorithm

---

1: **Input:** The cluster distributions $\{\nu_{\boldsymbol{a},c}\}$
2: **for all** $b \in \mathcal{B}$ **do**
3:      $\mathcal{C}_b \leftarrow \mathcal{C}$
4: **end for**
5: **for** $t = 1, \ldots, T$ **do**
6:      Receive $b \in \mathcal{B}$ and suppose this action was chosen at rounds $t_1, \ldots, t_n = t$
7:      Players construct $S_{\boldsymbol{a},b}(t)$ for all $\boldsymbol{a}$
8:      **if** $N_{\boldsymbol{a},b}(t) = 0$ **then**
9:          $S_{\boldsymbol{a},b}(t) \leftarrow (-\infty, \infty)$
10:      **end if**
11:      Select the next cluster $c(t_n)$ following $c(t_{n-1})$ in the ordered admissible set $\mathcal{C}_b$
12:      **if** $\mu_{\boldsymbol{a},c(t_n)} \notin S_{\boldsymbol{a},b}(t)$ for any $\boldsymbol{a}$ **then**
13:          $\mathcal{C}_b \leftarrow \mathcal{C}_b \setminus \{c(t_n)\}$
14:      **end if**
15:      Choose the next arm (break ties with ordering of arms)
16:      $\boldsymbol{a}_t \leftarrow \star_{c(t_n)}$
17: **end for**

---

The regret bound is as follows, with the proof deferred to the Appendix.

**Theorem 1** *The regret of Multiple-K-Intervals-A satisfies*

$$\mathcal{R}_{\mathcal{T}} \leq 8\sqrt{4T(\log T)}BC + 2MBC \tag{2}$$

Note a comparison with the regret bound of `Multiple-K-UCB` in Theorem 6 of Maillard & Mannor (2014). Their gap dependent bound contains a $\frac{\Delta_{\boldsymbol{a},c_b}}{(\Delta_{\boldsymbol{a},c_b}^+)^2}$ term, where $\Delta_{\boldsymbol{a},c}^+ = \inf_{c' \in \mathcal{C}} \left\{ \mu_{a,c'} - \mu_{\boldsymbol{a},c} : \star_{c'} = a \cap \mu_{\star_{c'},c'} \geqslant \mu_{\star_{\boldsymbol{c}},c} \right\}$. Since $\Delta_{\boldsymbol{a},c}^+ \geq \Delta_{\boldsymbol{a},c}$, their regret bound is bounded by $\frac{1}{\Delta_{\boldsymbol{a},c_b}^+}$, which essentially results in a gap-independent bound of the same order as the one presented above. However,

the proof technique we employ allows us to account for the adversarial nature's actions. Furthermore, a lower bound on the cumulative regret of any latent bandit algorithm, in the case $C > K^M$, is $\frac{1}{20}\sqrt{TK^M C}$, as shown in Maillard & Mannor (2014). Our bound is nearly optimal up to factor of logs.

### 3.1.1 Example

We now consider a setting to illustrate `Multiple-K-Intervals-A` in the case where there are three nature's actions and two clusters. Let nature's actions be denoted by $b_1, b_2, b_3$, with $b_1, b_3 \in \mathcal{B}_1$ (i.e., belonging to Cluster 1) and $b_2 \in \mathcal{B}_2$. Each cluster defines a reward distribution following Gaussian distributions $\mathcal{N}(\mu_{a,c}, 0.2^2)$ over joint actions $\boldsymbol{a} = (a_1, a_2) \in \{1, 2\} \times \{1, 2\}$.

Define $\mu_{\boldsymbol{a},c}$ and hence the cluster distributions as:

$$
\underbrace{\begin{array}{c} \phantom{1} \\ 1 \\ 2 \end{array}\begin{array}{cc} 1 & 2 \\ \left[\begin{array}{cc} 0.2 & 0.6 \\ 0.8 & 0.5 \end{array}\right] \end{array}}_{\text{Cluster 1}}, \underbrace{\begin{array}{c} \phantom{1} \\ 1 \\ 2 \end{array}\begin{array}{cc} 1 & 2 \\ \left[\begin{array}{cc} 0.8 & 0.1 \\ 0.7 & 0.3 \end{array}\right] \end{array}}_{\text{Cluster 2}}
$$

The rows represent Player 1's actions while the columns represent Player 2's actions.

Suppose the players agreed prior to learning to let the ordering of clusters be $(1, 2)$, and the order of joint arms be:

$$(1, 1) \to (1, 2) \to (2, 1) \to (2, 2) \tag{3}$$

Note that any order will suffice as long as the players use the same ordering. Initially, all clusters are admissible for all players under each state; as time progresses, players update their empirical means and confidence intervals for each joint arm under each state.

Suppose at round $t = 30$, the following occurs:

- Nature draws state $b_1 \in \mathcal{B}_1$ (true cluster is Cluster 1).

- The algorithm is currently testing Cluster 2 for $b_1$. Both players believe both clusters are admissible for $b_1$.

- For the optimal joint arm under Cluster 2, $(1, 1)$, the true mean under Cluster 2 is 0.8.

- Both players have pulled $(1, 1)$ many times under $b_1$ and observe (note that rewards are identical):

$$\text{Player 1: } \hat{\mu}^{(1)}_{(1,1),b_1} = 0.3, \quad S^{(1)} = [0.1, 0.5]$$

$$\text{Player 2: } \hat{\mu}^{(2)}_{(1,1),b_1} = 0.3, \quad S^{(2)} = [0.1, 0.5]$$

Since $\mu_{(1,1),2} = 0.8 \notin S^{(1)}$ and $\mu_{(1,1),2} \notin S^{(2)}$, both players simultaneously identify Cluster 2 as inadmissible under $b_1$. Both players update: $\mathcal{C}_{b_1} \leftarrow \mathcal{C}_{b_1} \setminus \{2\}$. If nature selects $b_1$ in the future, both players will aim for action $(2, 1)$, the optimal arm of the only admissible cluster for $b_1$. Later, nature selects $b_3$, which also belongs to $\mathcal{B}_1$. If both clusters are currently admissible for $b_3$, then players may still test Cluster 2 by pulling its optimal arm. Suppose nature selects $b_3$ again; then players will test the next admissible cluster, which is Cluster 1.

### 3.2 Problem B: Information Asymmetry in Rewards

The approach used in Problem A relies on all players observing identical rewards, allowing them to construct identical confidence intervals. This enables players to remain synchronized without needing to observe others' actions. In Problem B, however, players receive different rewards and maintain different confidence

intervals. Consequently, players may disagree on which clusters are admissible, creating the potential for miscoordination.

We address this by building on Algorithm 1 and introducing *sabotage,* in which individual players use observability of actions to signal that a cluster is no longer admissible based on their confidence intervals. The pseudocode is given in Algorithm 2, with the sabotage step in step 13.

**Intuition:** `Multiple-K-Intervals-B` builds on `Multiple-K-Intervals-A`, with one key difference to account for private rewards. Since players no longer maintain the same confidence intervals and admissible sets, they must implicitly communicate any belief changes. In `Multiple-K-Intervals-B`, agents sabotage by purposely deviating from the agreed-upon order when they observe that the cluster currently being tested, $c(t_n)$, is no longer admissible. This signals to other players that they should also remove that cluster from their admissible set for the current nature's action $b$.

The regret bound is as follows; the proof is deferred to the Appendix.

**Theorem 2** *The regret of Multiple-K-Intervals-B satisfies*

$$\mathcal{R}_T \leq 8\sqrt{4T(\log T)}BC + 2MBC + BC, \tag{4}$$

---

**Algorithm 2** The `Multiple-K-Intervals-B` Algorithm

1: **Input:** The cluster distributions $\{\nu_{\boldsymbol{a},c}\}$
2: **for all** $b \in \mathcal{B}$ **do**
3:     $\mathcal{C}_b \leftarrow \mathcal{C}$
4: **end for**
5: **for** $t = 1, \ldots, T$ **do**
6:     Receive $b \in \mathcal{B}$, and suppose this action was chosen at rounds $t_1, \ldots, t_n = t$
7:     Players construct $S_{\boldsymbol{a},b}^i(t)$ for all $\boldsymbol{a}$
8:     **if** $N_{\boldsymbol{a},b}(t) = 0$ **then**
9:         $S_{\boldsymbol{a},b}^i(t) \leftarrow (-\infty, \infty)$
10:     **end if**
11:     Select the next cluster $c(t_n)$ following $c(t_{n-1})$ in the ordered admissible set $\mathcal{C}_b$
12:     **if** player $i$ observes $\mu_{\boldsymbol{a},c(t_n)} \notin S_{\boldsymbol{a},b}^i(t)$ for some $\boldsymbol{a}$ **then**
13:         Player $i$ pulls a non-optimal arm to indicate to other players that $c(t_n)$ is no longer admissible for state $b$
14:         Each player observes the action and updates $\mathcal{C}_b \leftarrow \mathcal{C}_b \setminus \{c(t_n)\}$
15:     **end if**
16:     Choose the next arm:
17:     $\boldsymbol{a}_t \leftarrow \star_{c(t_n)}$
18: **end for**

---

### 3.2.1 Example

We now consider a setting to illustrate `Multiple-K-Intervals-B`, using the same setup as the example in section 3.1.1. Now suppose $b_3 \in \mathcal{B}_1$ and $b_1, b_2 \in \mathcal{B}_2$.

Define the cluster distributions as:

$$\begin{array}{cc} & \begin{array}{cc} 1 & 2 \end{array} \\ \begin{array}{c} 1 \\ 2 \end{array} & \underbrace{\left[\begin{array}{cc} 0.2 & 0.6 \\ 0.8 & 0.5 \end{array}\right]}_{\text{Cluster 1}} \end{array}, \begin{array}{cc} & \begin{array}{cc} 1 & 2 \end{array} \\ \begin{array}{c} 1 \\ 2 \end{array} & \underbrace{\left[\begin{array}{cc} 0.8 & 0.1 \\ 0.7 & 0.3 \end{array}\right]}_{\text{Cluster 2}} \end{array}$$

The rows represent Player 1's actions while the columns represent Player 2's actions. Again let the ordering of clusters be $(1, 2)$, and the order of joint arms be the same as above.

Suppose at round $t = 30$, the following occurs:

- Nature draws state $b_1 \in \mathcal{B}_1$ (true cluster is Cluster 1).

- The algorithm is currently testing Cluster 1 for $b_1$. Both players believe both clusters are admissible for $b_1$.

- For the joint arm $(2, 1)$, the true mean under Cluster 1 is 0.8.

- Both players have pulled $(2, 1)$ several times under $b_1$ and observe:

$$\text{Player 1: } \hat{\mu}^{(1)}_{(2,1),b_1} = 0.75, \quad S^{(1)} = [0.68, \ 0.82]$$

$$\text{Player 2: } \hat{\mu}^{(2)}_{(2,1),b_1} = 0.72, \quad S^{(2)} = [0.65, \ 0.79]$$

Since $\mu_{(2,1),1} = 0.8 \notin S^{(2)}$, Player 2 identifies Cluster 1 as inadmissible under $b_1$, while the interval for Player 1 still suggests both clusters are admissible. Player 2 deviates from the optimal arm of Cluster 1 to signal that it should be eliminated.

Player 1 observes this deviation, and both players update:

$$\mathcal{C}_{b_1} \leftarrow \mathcal{C}_{b_1} \setminus \{1\}$$

Later, suppose nature selects state $b_2$, which also belongs to $\mathcal{B}_2$. Since $\mathcal{C}_{b_2}$ has not yet been updated, players begin testing Cluster 1 again. Assume Player 1 observes Cluster 1 is no longer admissible according to their own confidence interval, but Player 2 does not. In that round: Player 1 deviates from the expected arm (the optimal arm of Cluster 1) to indicate Cluster 1 is inadmissible. Player 2 observes the deviation, and both players update $\mathcal{C}_{b_3} \leftarrow \mathcal{C}_{b_3} \setminus \{1\}$.

### 3.3 Problem C: Information Asymmetry in Both Rewards and Actions

In problem C, players can observe neither other players' rewards nor their actions, necessitating a different approach. Players maintain different confidence intervals, but cannot communicate changing beliefs to other players and may become miscoordinated.

To address this, we propose an explore-then-commit style algorithm, `Multiple-K-mDSEE`. In the basic `mDSEE` algorithm (Chang et al. (2022)), players alternate between exploration and commitment phases, spaced at increasing powers of 2. Players individually pick an action to commit to based on the empirical means observed during exploration phases. As the empirical means converge to the true means, commitment phases also increase in length; intuitively, this allows players to commit for longer periods once they are more confident and agree on the same optimal arm for each $b$.

**Intuition:** In our multiplayer latent bandit setting, we run `mDSEE` separately for each nature's action $b$, alternating between exploration and exploitation at predetermined intervals (with exploitation intervals increasing in length). We set a fixed exploration parameter $E$ based on gaps within clusters to ensure sufficient concentration of the empirical means. This allows us to improve on the results of Chang et al. (2022) in several ways: First, the fact that K depends on the phase $\lambda$ in Chang et al. (2022) is no longer necessary because we know the reward distributions. In this setting, we improve the order of regret in $T$ by utilizing knowledge of the reward gaps.

Note that players do not update their empirical means based on rewards observed during commitment phases; this is because players may commit to different arms during these phases but cannot observe the actions taken by other players, and thus do not know what joint action is taken. Additionally, players can restrict their search to $C$ arms, as they are given the optimal arm for each cluster and need not explore suboptimal arms.

The regret bound is as follows.

**Theorem 3** *The regret of Multiple-K-mDSEE satisfies*

$$\mathcal{R}_T \leq O\left(\frac{BC\log(T)}{\min_{a^\star,b}\Delta^2_{a^\star,b}} + BCM\frac{\pi^2}{3}\right).$$ (5)

We are able to show that the regret grows logarithmically with respect to a fixed set of reward gaps. The proof is deferred to the Appendix.

---

**Algorithm 3** `Multiple-K-mDSEE` Algorithm
___
1: **Input:** The cluster distributions $\nu_{a,c}$, "elite" arms $\mathscr{A}^\star = \{a^\star \mid \exists c \ a^\star = \arg\max_a \mu_{a,c}\}$, exploration parameter $E = 4/\left(\frac{1}{2}\min_{a^\star,b}\Delta_{a^\star,b}\right)^2$.
2: **Initialize:** $\lambda \leftarrow 1$.
3: **for** $t = 1, \ldots, T$ **do**
4:     Receive $b = b_t$.
5:     **if** $\exists a^\star \in \mathscr{A}^\star$ such that arm $a^\star$ has been pulled fewer than $E$ times in state $b$ for the $\lambda$th phase **then**
6:         Pull arm $a^\star$ in state $b$.
7:         Each player $i$ observes their own reward and updates $\widehat{\mu}^i_{a^\star,b}$.
8:     **else**
9:         Each player pulls their optimal arm $a^\star_t \leftarrow \arg\max_{a^\star}\widehat{\mu}^i_{a^\star,b}(t)$.
10:         Do not update $\widehat{\mu}^i_{a^\star,b}$.
11:         **if** $t + 1 = 2^n$ for some $n \geq \lambda$ **then**
12:             $\lambda \leftarrow \lambda + 1$.
13:         **end if**
14:     **end if**
15: **end for**

---

## 4 Experiments

We empirically evaluate the three algorithms in the multi-agent multi-armed bandit setting with nature's actions. The algorithms differ in how players handle uncertainty in state-cluster mappings and how they coordinate under information asymmetry.

**Setup:** We ran each algorithm under 20 randomly initialized environments and computed the average cumulative regret at $t = T$. Each nature's action $b \in \{0, 1, \ldots, B-1\}$ is deterministically assigned to a cluster $c_b \in \{0, 1, \ldots, C-1\}$ using the rule:

$$c_b = \min\left(\left\lfloor \frac{b}{\lfloor B/C \rfloor}\right\rfloor, C-1\right)$$

The reward for each arm-state pair is sampled from a Gaussian distribution $\mathcal{N}(\mu_{a,c}, 0.4^2)$, where the cluster reward tables $\{\mu_{a,c}\}_{a \in [K]^M, c \in [C]}$ are initialized randomly from a uniform distribution at the start of each trial. Each table $\mu^{(c)}$ corresponds to a specific cluster $c$ and defines the mean reward for every joint arm $a = (a_1, \ldots, a_M)$ played by the $M$ players.

**Results:** Figure 1 shows the cumulative regret averaged over 20 trials for each of the three algorithms evaluated. The red curve, `Multiple-K-UCB` from Maillard & Mannor (2014) adapted for the multiplayer joint-action problem, achieves sublinear regret performance across the horizon as expected. This algorithm can only be run for Problem A, as it requires visibility of rewards to be applied in the multiplayer case.

The blue curve, our proposed algorithm `Multiple-K-Intervals-B` for Problem B, demonstrates performance of a similar order to `Multiple-K-UCB` without requiring observability of rewards. Note that `Multiple-K-Intervals-A`'s bound is off by only a small constant, and is not plotted separately here.

The green curve, `Multiple-K-mDSEE`, incurs significantly higher cumulative regret as Problem C is much more difficult owing to observability of neither actions nor rewards. The "staircase" pattern is a result of

periodic resets in the explore-then-commit cycles of `Multiple-K-mDSEE`: At certain points, increasing the horizon $T$ allows the algorithm to enter another exploration phase, hence the jump in regret. Exploration phases are spaced further and further apart, causing the flat sections to increase exponentially in length. While `Multiple-K-mDSEE` can be applied to Problems A and B, it performs much worse, which is to be expected due to its information-constrained nature.

The orange curve represents a naive UCB approach, in which UCB is run "in parallel" for each nature's action $b$ (where $\hat{\mu}(t)$ and $N(t)$ are tracked separately for each $b$.) Note also that this naive approach is only possible for Problem A, in which players can stay synchronized because they receive identical rewards and can infer each others' actions; this algorithm cannot be applied directly to Problem B or C. The naive UCB algorithm has poor performance relative to the other algorithms, demonstrating the value in utilizing the latent structure of the problem.

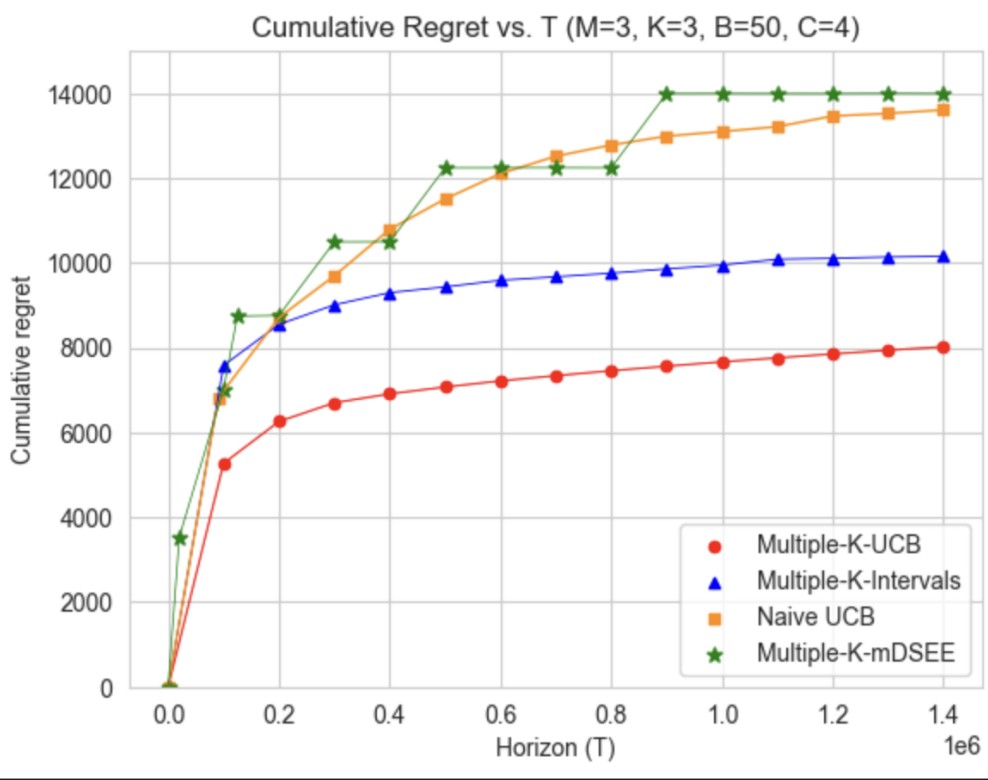

Figure 1: Cumulative regret over values of $T$ from 100000 to 1400000, with $M = 3$ players, $K = 3$ arms, $C = 4$ clusters, the exploration parameter $E = 250$, and $B = 50$ nature's actions.

## 5 Conclusion

In this paper, we extended the latent multi-armed bandit framework to the novel multiplayer setting with information asymmetry and limited communication. We introduced three algorithms tailored to distinct asymmetry scenarios: `Multiple-K-Intervals-A` for shared rewards with unobservable actions, `Multiple-K-Intervals-B` for observable actions but private i.i.d. rewards, and `Multiple-K-mDSEE` for the case where neither rewards nor actions are shared. We verify our results through numerical experiments. Our results show that for Problems A and B, we achieve gap-independent regret bounds of order $O(\sqrt{T \log T} BC)$ while still allowing for nature's actions to be adversarial, improving on the original single-agent analysis in Maillard & Mannor (2014). For Problem C, we utilize knowledge of cluster structures to design an explore-then-commit strategy that results in a regret bound of order $O(BC \log(T)/(\min_{a^\star, b} \Delta_{a^\star, b})^2)$, improving on the results in Chang et al. (2022).

However, several directions for future work remain. The regret bound for problem C scales on the minimum gap between potentially optimal arms, possibly leading to poor performance in certain cases. Future work could explore gap-free strategies for a tighter bound on regret. In addition, our algorithms rely on known reward distributions for each cluster, which may not be realistic in all applications. Investigating multiplayer algorithms for unknown cluster distributions (the agnostic case from Maillard & Mannor (2014)) remains an open challenge.

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

## A  Important Lemmas

**Lemma 4 (corollary 5.5 of Lattimore & Szepesvári (2020))** *Assume that $X_i - \mu$ are independent, $\sigma$-subgaussian random variables. Then for any $\varepsilon \geq 0$,*

$$\mathbb{P}(\hat{\mu} \geq \mu + \varepsilon) \leq \exp\left(-\frac{n\varepsilon^2}{2\sigma^2}\right) \quad and \quad \mathbb{P}(\hat{\mu} \leq \mu - \varepsilon) \leq \exp\left(-\frac{n\varepsilon^2}{2\sigma^2}\right),$$

*where $\hat{\mu} = \frac{1}{n}\sum_{t=1}^{n} X_t$.*

**Lemma 5** *Define the good event across all players, nature's actions, and clusters as*

$$G = \bigcap_{t=1}^{T}\bigcap_{i=1}^{M}\bigcap_{b\in\mathcal{B}}\bigcap_{c\in\mathcal{C}} G^i_{\star_c,b}(t), \quad G^i_{\boldsymbol{a},b}(t) = \{\mu_{\boldsymbol{a},b} \in S^i_{\boldsymbol{a},b}(t)\}$$

*where $S^i_{\boldsymbol{a},b}(t) = \left(\hat{\mu}^i_{\boldsymbol{a},b}(t) - \sqrt{\frac{4\log(T)}{N^i_{\boldsymbol{a},b}(t)}}, \hat{\mu}^i_{\boldsymbol{a},b}(t) + \sqrt{\frac{4\log(T)}{N^i_{\boldsymbol{a},b}(t)}}\right).$*

*We have the following upper bound*

$$P(G^c) \leq \frac{2MBC}{T}. \tag{6}$$

**Proof:**  By DeMorgan's rule, taking the complement of the good event we have

$$G^c = \bigcup_{t=1}^{T}\bigcup_{i}\bigcup_{b}\bigcup_{c}\{\mu^i_{\star_c,b} \leq L^i_{\star_c,b}(t) \text{ or } \mu^i_{\star_c,b} \geq U^i_{\star_c,b}(t)\}$$

$$= \bigcup_{t=1}^{T}\bigcup_{i}\bigcup_{b}\bigcup_{c}\{\mu^i_{\star_c,b} \leq L^i_{\star_c,b}(t)\} \cup \{\mu^i_{\star_c,b} \geq U^i_{\star_c,b}(t)\}.$$

By the probability union bound, we have

$$P(G^c) \leq \sum_{t=1}^{T}\sum_{i}\sum_{b}\sum_{c} P(\mu^i_{\star_c,b} \leq L^i_{\star_c,b}(t)) + P(\mu^i_{\star_c,b} \geq U^i_{a,b}(t)). \tag{7}$$

By Lemma 4, we upper bound

$$P(\mu_{\star_c,b} \leq L^i_{\star_c,b}(t)) = P\left(\mu_{\star_c,b} \leq \hat{\mu}^i_{\star_c,b} - \sqrt{\frac{4\log(T)}{N^i_{\star_c,b}(t)}}\right)$$

$$\leq \exp\left(-\frac{N^i_{\star_c,b}(t)\left(\sqrt{\frac{4\log(T)}{N^i_{\star_c,b}(t)}}\right)^2}{2}\right)$$

$$= \exp(-2\log(T))$$

$$= \frac{1}{T^2}.$$

Now we plug this into our upper bound for $P(G^c)$:

$$P(G^c) \leq \sum_{t=1}^{T}\sum_{i}\sum_{b}\sum_{c}\frac{2}{T^2} = \frac{2MBC}{T}. \tag{8}$$

$\square$

## B  Regret bound of `Multiple-K-Intervals-A`

We present the proof of Theorem 1, the regret bound of `Multiple-K-Intervals-A` for Problem A, information asymmetry in actions.

**Proof:**  Consider the good event

$$G = \bigcap_{t=1}^{T} \bigcap_{i=1}^{M} \bigcap_{b \in \mathcal{B}} \bigcap_{c \in \mathcal{C}} G_{\star_c, b}^i(t), \quad G_{\boldsymbol{a}, b}^i(t) = \{\mu_{\boldsymbol{a}, b} \in S_{\boldsymbol{a}, b}^i(t)\}, \tag{9}$$

where $S_{\boldsymbol{a}, b}^i(t)$ is the confidence interval of player $i$ at time $t$. Recall the definition of regret

$$\mathcal{R}_T = \sum_{t=1}^{T} \mathbb{E}[X_{\star_{\boldsymbol{b}_t}, b_t} - X_{\boldsymbol{a}_t, b_t}] \tag{10}$$

$$= \sum_{t=1}^{T} \mathbb{E}[X_{\star_{\boldsymbol{b}_t}, b_t} - X_{\boldsymbol{a}_t, b_t} | G] P(G) + \mathbb{E}[X_{\star_{\boldsymbol{b}_t}, b_t} - X_{\boldsymbol{a}_t, b_t} | G^c] P(G^c) \tag{11}$$

$$\leq \sum_{t=1}^{T} \mathbb{E}[X_{\star_{\boldsymbol{b}_t}, b_t} - X_{\boldsymbol{a}_t, b_t} | G] + T P(G^c). \tag{12}$$

Denote

$$\delta_{\boldsymbol{a}, b}(t) = U_{\boldsymbol{a}, b}(t) - L_{\boldsymbol{a}, b}(t) = 2\sqrt{\frac{4 \log T}{N_{\boldsymbol{a}, b}(t)}}.$$

Suppose that at time $t$, nature selects an action $b_t$ belonging to cluster $c_t$. At each round, the players select the best arm for a cluster that is admissible. More explicitly, $\boldsymbol{a_t} = \star_{\boldsymbol{c}'}$, where $c'$ is admissible. Since $c'$ is admissible, $\mu_{\star_{\boldsymbol{c}_t}, c'} \in S_{\star_{\boldsymbol{c}_t}, b_t}(t)$. We also assume that the true cluster $c_t$ is admissible under the good event, thus $\mu_{\star_{\boldsymbol{c}_t}, c_t} \in S_{\star_{\boldsymbol{c}_t}, b_t}(t)$, and thus

$$\mu_{\star_{\boldsymbol{c}_t}, c_t} - \mu_{\star_{\boldsymbol{c}_t}, c'} \leq \delta_{\star_{\boldsymbol{c}_t}, b_t}(t).$$

Similarly, since $\mu_{\star_{\boldsymbol{c}'}, c'} \in S_{\star_{\boldsymbol{c}'}, b_t}(t)$, we have

$$\mu_{\star_{\boldsymbol{c}'}, c'} - \mu_{\star_{\boldsymbol{c}'}, c_t} \leq \delta_{\star_{\boldsymbol{c}'}, b_t}(t).$$

Further, $\mu_{\star_{\boldsymbol{c}_t}, c'} - \mu_{\star_{\boldsymbol{c}'}, c'} \leq 0$, since $\star_{c'}$ is the optimal arm for cluster $c'$; recall that $\mu_{\boldsymbol{a}, c}$ is the true mean of arm $\boldsymbol{a}$ under cluster $c$. Thus

$$\sum_{t=1}^{T} \mathbb{E}[X_{\star_{b_t},b_t} - X_{a_t,b_t}|G] = \sum_{t=1}^{T} \left(\mu_{\star_{c_t},c_t} - \mu_{\star_{c'},c_t}\right) \tag{13}$$

$$= \sum_{t=1}^{T} \left((\mu_{\star_{c_t},c_t} - \mu_{\star_{c_t},c'}) + (\mu_{\star_{c_t},c'} - \mu_{\star_{c'},c'}) + (\mu_{\star_{c'},c'} - \mu_{\star_{c'},c_t})\right) \tag{14}$$

$$\leq \sum_{t=1}^{T} \left(\delta_{\star_{c_t},b_t}(t) + \delta_{\star_{c'},b_t}(t)\right) \tag{15}$$

$$= \sum_{t=1}^{T} \left(\delta_{\star_{c_t},b_t}(t) + \delta_{a_t,b_t}(t)\right) \tag{16}$$

$$= 2\sum_{t=1}^{T} \left(\sqrt{\frac{4\log T}{N_{\star_{c_t},b}(t)}} + \sqrt{\frac{4\log T}{N_{a_t,b}(t)}}\right) \tag{17}$$

$$\leq 2\sum_{b\in\mathcal{B}}\sum_{c\in\mathcal{C}}\sum_{s=1}^{N_{a,b}(T)} 2\sqrt{\frac{4\log T}{s}} \tag{18}$$

$$\leq 4\sqrt{4\log T}\sum_{b}\sum_{c}\sum_{s=1}^{T}\frac{1}{\sqrt{s}} \tag{19}$$

$$\leq 4\sqrt{4\log T}\sum_{b}\sum_{c}\int_{0}^{T}\frac{1}{\sqrt{s}}ds \tag{20}$$

$$= 8\sqrt{4\log T}\sum_{b}\sum_{c}\sqrt{T} \tag{21}$$

$$= 8\sqrt{4\log T}BC\sqrt{T}. \tag{22}$$

Note that in equation (18) we take a summation across the clusters rather than the set of all arms, because for each nature's action $b$ we try at most $C$ arms (the optimal arms in each cluster).

Thus, plugging in the above result to 12 and combining with the bound for $P(G^C)$ from A:

$$\mathcal{R}_T \leq 8\sqrt{4\log T}BC\sqrt{T} + 2MBC. \tag{23}$$

$\square$

## C   Regret bound of Multiple-K-Intervals-B

We present the proof of Theorem 2, the regret bound of `Multiple-K-Intervals-B` for Problem B, information asymmetry in rewards.

**Proof:**   The only deviation from Algorithm 1 is the addition of sabotage. The cumulative regret incurred by sabotage across all rounds is at most $N_s \leq BC$. Thus $\mathcal{R}_T \leq 8\sqrt{4T(\log T)}BC + 2MBC + BC$.   $\square$

## D   Regret bound of Multiple-K-mDSEE

We present the proof of Theorem 3, the regret bound of `Multiple-K-mDSEE` for Problem C, information asymmetry in actions and rewards.

**Proof:**   Decompose $\mathcal{R}_T = \mathcal{R}_{T,E} + \mathcal{R}_{T,C}$, where $\mathcal{R}_{T,E}$ is the regret incurred from the exploration phases spaced at powers of 2, and $\mathcal{R}_{T,C}$ is the regret incurred from all commitment phases.

Thus, recalling that we only explore $\mathscr{A}^\star$, the set of arms that are optimal for some cluster, the regret incurred during exploration is at most:

$$\mathcal{R}_{T,E} \leq \sum_{b \in \mathcal{B}} \sum_{\boldsymbol{a}^\star \in \mathscr{A}^\star} E \lceil \log_2(T) \rceil \Delta_{\boldsymbol{a}^\star, b}, \tag{24}$$

where $\Delta_{\boldsymbol{a},b} = \mu_{\star_b, b} - \mu_{\boldsymbol{a},b}$ is the suboptimality gap for arm $\boldsymbol{a}$ under state $b$ and $E = \dfrac{4}{\left(\frac{1}{2} \min_{\boldsymbol{a}^\star, b} \Delta_{\boldsymbol{a}^\star, b}\right)^2}$.

Note that all unions, sums, and minimums in this proof relating to player actions $\boldsymbol{a}$ are taken over $\mathscr{A}^\star$, rather than the complete set $\mathscr{A}$.

We now analyze the **commitment phase** regret $\mathcal{R}_{T,C}$.

Let $\epsilon = \frac{1}{2} \min_{\boldsymbol{a}^\star, b} \Delta_{\boldsymbol{a}^\star, b}$.

In the latent multi-armed bandit setting with multiple-cluster arrivals, the sub-optimality gaps within clusters are known due to the known reward structure. This allows us to choose a fixed exploration constant $E$ large enough to ensure sufficient concentration of the empirical means. If each arm is pulled $E$ times during the previous exploration phases, then we have the following inequality when $t$ is in the committing phase:

$$N_{\boldsymbol{a}^\star, b}(t) \geq E \log_2(t). \tag{25}$$

Define the good event for an individual player $i$:

$$G^i_{\boldsymbol{a}^\star, b}(t) = \left\{ \left| \hat{\mu}^i_{\boldsymbol{a}^\star, b}(t) - \mu_{\boldsymbol{a}^\star, b} \right| < \epsilon \right\}. \tag{26}$$

When $\bigcap_b \bigcap_{\boldsymbol{a}^\star} \bigcap_i G^i_{\boldsymbol{a}, b}(t)$ occurs, the optimal arm is selected at round $t$, and regret is 0. Thus,

$$\mathcal{R}_{T,C} \leq \sum_{b \in \mathcal{B}} \sum_{t: b_t = b} \mathbb{E}\left[ \mathbb{I}\left[ \left( \bigcap_{\boldsymbol{a}^\star} \bigcap_{i=1}^{M} G^i_{\boldsymbol{a}, b}(t) \right)^c \right] \right] \tag{27}$$

$$= \sum_b \sum_{t: b_t = b} \mathbb{P}\left[ \left( \bigcap_{\boldsymbol{a}^\star} \bigcap_i G^i_{\boldsymbol{a}, b}(t) \right)^c \right] \tag{28}$$

$$\leq \sum_b \sum_{t=1}^{n_b(T)} \sum_{\boldsymbol{a}^\star} \sum_i \mathbb{P}\left[ G^i_{\boldsymbol{a}, b}(t)^c \right] \tag{29}$$

$$= \sum_b \sum_{t=1}^{n_b(T)} \sum_{\boldsymbol{a}^\star} \sum_i \mathbb{P}\left[ \left| \hat{\mu}^i_{\boldsymbol{a}, b}(t) - \mu_{\boldsymbol{a}, b} \right| > \epsilon \right] \tag{30}$$

$$\leq \sum_b \sum_{t=1}^{n_b(T)} \sum_{\boldsymbol{a}^\star} \sum_i 2 e^{-\frac{n_{\boldsymbol{a}, b}(t) \epsilon^2}{2}} \tag{31}$$

$$\leq \sum_b \sum_{\boldsymbol{a}^\star} \sum_{t=1}^{n_b(T)} M \cdot 2 e^{-\frac{E \log_2(t) \epsilon^2}{2}} \tag{32}$$

$$\leq \sum_b \sum_{\boldsymbol{a}^\star} \sum_{t=1}^{n_b(T)} M \cdot 2 t^{-2} \tag{33}$$

$$\leq BCM \frac{\pi^2}{3}. \tag{34}$$

In the second inequality we use the probability union bound and in the third inequality we use the fact that the rewards are i.i.d., so the probability of the complement of the good event has the same upper bound

for each player. In the last inequality, we use the fact that $|\mathscr{A}^\star| \leq C$, assuming unique optimal actions; the proof is easily generalized for non-unique optimal actions. Note that unlike in Chang et al. (2022), we can define $\epsilon$ explicitly, leveraging knowledge of the gaps, rather than fixing an exploration schedule to guarantee theoretical convergence.

Thus our total regret $\mathcal{R}_T$ satisfies $O(BC \log(T)/(\min_{a^\star,b} \Delta_{a^\star,b})^2 + BCM\frac{\pi^2}{3})$.

$\square$

