# OpenReview forum: "Adversarially Robust Latent Bandits in Multiplayer Asymmetric Settings"
_TMLR — Rejected by TMLR_

### Review · Reviewer_HbgZ · 2026-02-22

**Summary Of Contributions:**

This paper studies multiplayer extension of latent multi-armed bandits under the information asymmetry setup. Building upon the latent bandit formulation of Maillard & Mannor (2014) and the multiplayer asymmetric framework of Chang et al. (2022), the authors analyze three observation models, which differ in the observability of the actions and the assumptions on the rewards. They assume the rewards of the clusters are given, and the learning problem reduces to a cluster identification problem. Three cluster elimination algorithms, based on the confidence intervals or the explore-then-commit scheme, are given, as well as regret analysis for the three models.

**Additional Comments:**

- The technical contribution is incremental. The techniques, including the confidence-interval elimination, the mdSEE-style exploration and the deliberate deviation, are standard in the literature. The novelty of this paper mostly lies in combining these techniques.
- The assumptions, (1) the cluster reward distributions and the optimal arm are given, and (2) the reward distributions are identical within each cluster, are quite strong. In the practical scenarios, this information is usually not revealed and also requires inference from the observations.
- The adversarial robustness claim is only in terms of the nature’s action, i.e., the choice of the states. As the agents cannot control the nature, but can observe the state, it seems the model is close to the contextual setup. It is appreciated that the author can compare the adversarial setup and the contextual setup.
- For problem B, the regret bound contains $N_s$, the number of rounds in which any player selects a suboptimal action to sabotage. This (random) quantity depends on the algorithm, and should be further bounded.
- For the experiments, Figure 1 is misleading. In the caption, it says it is the plot for a fixed $T=10^6$. Therefore, it is expected that there should be 1 datapoint for each method, where as there a curve for each method for $T\in(1,10^6)$. If this is the plot for the regret accumulated at $t\in  (1,10^6)$ with time horizon $T=10^6$, then it cannot support any theoretical claim in the previous sections, where the guarantees are in terms of the whole time horizon $T$.
- The experiment is weak, as it only tested Gaussian rewards with small variance and the number of test trials is small.
- The lower bound for each of the three problems are not given.

**Audience:**

Yes

**Audience Explanation:**

- The proposed formulation is interesting. It combines the latent bandit and the multiple player coordination together, which is well-motivated.
- It allows the states to be adversarially chosen by the nature.

**Claims And Evidence:**

Yes

**Claims Explanation:**

- Each problem is well elaborated and theorems are given to summarize the results.
- However, Figure 1 looks misleading. Please kindly see the additional comments section.

**Requested Changes:**

Please also kindly see the additional comments section.

- Please clarify Figure 1.
- It is appreciated that if the lower bounds for the three setups are given. As there are much more information is given, including the cluster distributions and the optimal arm, it is expected that less regret can be achieved.

---

> ### Author Response · Authors · 2026-03-05
>
> Thank you for your valuable feedback and careful review of our work. We address each point below:
>
> **Requested Changes:**
> 1. Figure 1: We agree that the figure is confusing as is. We have updated the experiments section to better support our theoretical results; in particular, we now show cumulative regret at $t=T$ for different values of $T$ for each algorithm to demonstrate that dependency on $T$ is as expected (averaged across all trials). We will also include experiments with less well-behaved environments in the final version, particularly with larger variance in rewards.
> 2. Lower bounds: We cite the lower-bound template from [1] in the discussion after Theorem 1; we will make this more explicit for our multi-agent setting (using $K^M$ instead of $K$). We agree that less regret is achievable than in the fully agnostic setting, since we know the cluster structures. In general, since cluster distributions and optimal arms are given, we only need to explore the arms that are known to be optimal for some cluster; this is reflected by the presence of $C$ instead of $K^M$ in our bounds.
>
> **Additional comments:**
> 1. While we agree that some elements of our analyses are standard, we want to emphasize the areas of novelty below:
>
> a. Our formulation of the latent problem, with joint actions, under 3 types of information asymmetry, is novel.
>
> b. We derive gap-independent bounds for Problems A and B without making any assumptions on the distribution of nature's actions, which is nonstandard among latent bandit works.
>
> c. Our version of mDSEE is nonstandard in that it leverages the known structure in the latent problem to form a gap-informed exploration schedule.
> We have also expanded the Related Works section to make our contribution more clear.
>
> 2. Our formulation is designed for applications where there is offline data available to estimate cluster structure from. Indeed, this requires using some clustering algorithm to estimate clusters, which allows for faster learning than if the latent structure is ignored. For instance, recommendation systems can estimate clusters based on past data and then assign new users to a cluster rather than learning from scratch.
> 3. You are correct that only nature acts adversarially. We have revised the Related Works section to include a discussion of the difference between the contextual and latent problems. Mostly, the difference is that we do not assume that rewards and context/action are linked by a parametric model class. In contrast, in the latent problem we are primarily interested in the cluster partitions.
> 4. In the proof sketch we mention that $N_s \\leq BC.$ We have moved this into the main theorem statement to make this more explicit.
>
> 5, 6, 7. Please refer to our responses under Requested Changes.
>
> [1] Maillard, Odalric-Ambrym, and Shie Mannor. "Latent Bandits." International Conference on Machine Learning. PMLR, 2014.

---

### Review · Reviewer_hLQg · 2026-02-24

**Summary Of Contributions:**

This paper studies the multiplayer bandit problem with latent variables that determine the cluster assignment of players. In particular, it investigates the impact of information asymmetry and adversarial actions on the regret. To address this setting, the paper considers three scenarios that differ in whether the reward and action sequences are observable. Corresponding to these scenarios, three algorithms are proposed, each accompanied by theoretical regret guarantees. Finally, the paper evaluates the numerical performance of the proposed methods through simulation experiments.

**Additional Comments:**

N/A

**Audience:**

Yes

**Audience Explanation:**

This paper is relevant to the audience for several reasons:

1. The multiplayer bandit problem is widely studied and has broad applications in real-world decision-making problems.

2. The information-asymmetric setting is closely related to the field of information theory.

3. The existence of adversary is an important topic in the bandit literature.

**Claims And Evidence:**

Yes

**Claims Explanation:**

All theorems and results are accompanied by the corresponding proof steps in Appendix.

**Requested Changes:**

1. My primary concern is that the concept of ``adversarial actions'' is not mathematically defined, even though it appears to be one of the key contributions of the paper. While one can qualitatively infer its meaning from the context—perhaps as arbitrary changes or choices—it remains unclear what the paper refers to in precise quantitative terms. In fact, the adversary model can vary significantly depending on the setting. This ambiguity raises concerns regarding the rigor of the formulation, as well as the clarity and validity of the stated regret bounds.

2. Another concern is that the proof steps appear to closely follow classical bandit analyses, if I am interpreting them correctly. Could the authors clarify and explicitly highlight the differences compared to standard proofs? Doing so would help better articulate the technical novelty of the work.

3. The related work section provides a useful overview of the relevant literature. However, for each line of work discussed, there is limited direct comparison with the present paper. Explicitly emphasizing the key distinctions would help readers better appreciate the specific contributions and advances made in this work.

4. How does this work compare to the line of research on multiplayer bandits with adversarial rewards? It would be helpful to include a more explicit discussion of the similarities and differences in the related work section.

5. I am confused by the statement in Section~2.2 (second paragraph) that rewards ``come from any of several clusters with known reward distributions.'' If the reward distributions are known, where does the learning problem arise? It is possible that I have misunderstood the intended meaning, but clarification would be greatly appreciated.

6. The regret bound in Section~3.3 depends on $K^M$, which grows exponentially in $M$. Could the authors comment on the implications of this dependence, particularly in terms of scalability?

7. In the experiments, UCB appears to outperform the proposed method up to $2 \times 10^5$ time steps. While it is understandable that the proposed algorithm may require time to demonstrate its advantages, this convergence speed warrants further justification. A brief discussion clarifying this phenomenon would strengthen the empirical section.

8. Additionally, the performance of ``Multiple-K-mDSE'' does not appear to significantly improve upon UCB. This may raise questions about the practical advantage or superiority of the proposed algorithm. Further clarification or analysis would help address this concern.

---

> ### Author Response · Authors · 2026-03-05
>
> We sincerely thank the reviewer for their thoughtful comments and review of our work. We address each point below:
>
> **Requested Changes:**
> 1. In our algorithms and analysis, nature's actions are adversarial, i.e. nature can select an arbitrary sequence of states that need not follow a stochastic distribution. Since we do not assume a fixed distribution for nature, no strict definition is required. The rewards conditional on nature's action, however, are stochastic. Our bounds for Problems A and B are worst-case over all possible state sequences $b_t.$
> 2. Thank you for raising this point; we will revise the proof sketches to highlight the technical novelty, which is mostly in the following areas:
>
> a) Our analyses do not make assumptions on the distribution of nature's actions, unlike other latent bandit works; thus, we actually improve the setting in [1] even in the single-player case.
>
> b) Our analysis for problem C holds despite asymmetric observations. Rather than using collisions or other forms of signaling as is standard, we utilize unique aspects of the latent bandit framework to ensure coordination between players, departing from the non-latent mDSEE analysis.
>
> 3. Thank you for pointing this out; please see the revised manuscript for a clearer discussion of related works.
> 4. While nature acts adversarially, our rewards conditioned on a given $b_t$ are stochastic, unlike in the adversarial problem where the rewards/losses themselves are adversarial. We have added a discussion of this in paragraph 6 of the Related Works to make the distinction more explicit.
> 5. For each cluster $c,$ we are given the mean of each arm. However, learners do not know which cluster each nature's action belongs to, thus the learning problem is to learn these partitions effectively.
> 6. With some further analysis, we are actually able to show that since players only ever explore arms that are optimal for some cluster (and there are at most $C$ of these), we can replace the $K^M$s in the bounds with $C,$ which scales much better. We have updated the proofs accordingly.
> 7. We agree the experimental section can be expanded. As discussed below, however, the UCB baseline plotted is only applicable to Problem A. The superiority of UCB can be attributed to the fact that it uses an easier information model, and it cannot serve as a baseline for Problems B and C. The algorithms for Problem B (multiple-K-Intervals) and Problem C (Multiple-K-mDSEE) pay heavier exploration costs to maintain coordination under limited observability.
> 8. In our experiments, Multiple-K-mDSEE cannot be compared to Naive UCB since Naive UCB can only be applied in Problem A, whereas mDSEE can be applied in all three problems with no observability requirements. While mDSEE is worse for some values of $T$, this is to be expected as it builds in more exploration in exchange for coordination.
>
> [1] Maillard, Odalric-Ambrym, and Shie Mannor. "Latent Bandits." International Conference on Machine Learning. PMLR, 2014.

---

### Review · Reviewer_2RHz · 2026-02-26

**Summary Of Contributions:**

The paper builds from the work of Chang et al. (2022) and Chang & Lu (2023) – which develop a framework for multi-player bandits with three information sharing settings pertaining to actions and rewards. Further recent work, Chang & Lu (2025) and Chang & Karthik (2025) extended this framework to contextual and metric bandits respectively. The present paper adds a further such extension to the literature, building on the latent bandit model of Maillard and Mannor (2014).

The multi-player latent bandit model supposes that M players have access to a discrete set of K arms/actions, nature chooses contexts from a discrete set of size B, and elements of B are divided into C<B clusters. In each round a context is drawn and the players (independently) select actions forming an action vector a in K^M. The reward generated is a draw from a distribution $\nu_{a,b}$ which is identical for all b within the same cluster. The setup supposes players know the mean reward per-action-vector-per-cluster, but do not know which contexts belong to which cluster, nor can they communicate fully with the other players, during the game. The rewards are drawn stochastically, the ‘adversarial’ in the paper’s title refers to adversarial selection of the contexts.

Three informational settings consider the sharing of action and reward information and the authors propose a method, building on the approach from Maillard and Mannor for each. Regret bounds are derived for all three problems (gap-independent for Problems A and B, and gap-dependent for Problem C where there is the least information sharing). I think it is fair to say that the regret analysis is an application and slight extension of standard regret analysis techniques. A numerical experiment on a single problem instance compares the proposed algorithms to a naïve UCB based approach.

References
-	Maillard and Mannor (2014). Latent Bandits.
-	Chang, Jafarnia-Jahromi, and Jain (2022). Online learning for cooperative multi-player multi-armed bandits.
-	Chang and Lu (2023). Optimal cooperative multiplayer learning bandits with noisy rewards and no communication.
-	Chang and Karthik (2025). Multiplayer information asymmetric bandits in metric spaces.
-	Chang and Lu (2025). Multiplayer information asymmetric contextual bandits.

**Audience:**

Yes

**Audience Explanation:**

The paper is in scope and combines two aspects of bandit problems (latent cluster structure, and multi-player actions) that have not been combined in this way before.

**Broader Impact Concerns:**

I don't have any concerns for this work.

**Claims And Evidence:**

Yes

**Claims Explanation:**

Yes, besides some points of confusion I note below. The results themselves are clearly presented and justified.

**Requested Changes:**

I have a number of 'major' change areas where I feel that I need a response from the authors to approach any acceptance recommendation, and 'minor' changes which are not critical to my overall evaluation, but I think are mostly objective areas for refinement.

Major Comments

-	Related Work: The related literature section could do more to serve the motivation of the work and raises some questions as to whether state-of-the-art methodologies are being used. The authors base their work on the model and algorithms of Maillard and Mannor (2014) but note in the third and fourth paragraph of the Related Works setting a wide range of more recent papers developing methods for latent bandits. What is the rationale for not utilising these here? In a similar vein, accounts are given of competing and cooperative bandits and competitive RL, but the section needs revision to justify where the connections between these models and the framework of Chang et al. (2022) lie, or do not. This is important to verify the novelty of the work.

-	Inconsistencies and typos in Proofs:

o	In the Proof of Lemma 5, \hat{\mu}_{a,b}, \mu_{a,b}, and N_{a,b}(t) appear and disappear for two lines – are these a typo, or is there some explanation missing?

o	The good event in (9) is posed in terms of i-dependent S, U, etc. – why does everything in Lemma 5’s proof ignore i-dependence without explanation?

o	The statement of Lemma 5 has minimal detail – it is assumed the reader can infer that this is in the context of Algorithms 1 and 2.

o	It would be clearer if the ranges for b, c, and i were provided when they index unions or sums – at least in the first line of the proofs where they appear.

o	On p15, you aim to show the regret is less than a sum of the delta terms – is this not what you aim to show the sum of regrets conditioned on the good event is less than? The regret also includes the 2BC term derived from the good event not holding?

o	Equation (15) has no associated textual explanation but quite critical. With the fairly complex dependence on clusters etc it would be worth supporting the reader here.


-	Proof for Problem B: The bound for Problem B seems to be left in terms of a random variable N_s – the number of sabotages. Ideally we would have a constant (for fixed T) as the bound. However, the proof for Problem B is stated to be effectively the same as Problem A, except for the addition of sabotage. What confuses me is that sabotage actions are only triggered when the good event does not hold (see line 12 Alg 2), if I have understood correctly. There is already a worst-case assumption on the regret when the good event does not hold -  so does sabotage need to be captured at all?
-	Tightness of Bounds: The note after Theorem 1 makes a comparison to the lower bound from Maillard and Mannor, stating that the result is nearly optimal up to a factor of logs -  this seems to ignore the different dependence on B and C? If C is indeed > K^M this would seem to be an important factor here? Is this worth commenting on? For instance, although the bounds match in order up to logarithmic terms, in the problem considered in the experiment, it appears that the bound from theorem 1, 8BC(4Tlog(T))^0.5 + 2BC, is only < T once T reaches around 1.9x10^8. At T=10^6, the horizon of the experiments, the bound has value 1,1894,551 compared to an average regret of 6,000 to 12,000 for the algorithms considered. To me, this slackness is a limiting factor - meaning that the methodology and the empirical results are more valuable than the theory - unless there are ways to tighten the analysis?


Minor Comments

-	Some of your references are not in parentheses that should be (e.g. throughout paragraph 2)

-	Paragraph 2 describes the broad area of multi-agent RL but it doesn’t really connect to bandits as a specific instance within that. I would have expected a summary of multi-player bandits here rather than RL at large.

-	Many of the references refer to arxiv preprints rather than the published versions of the work.

-	Some details can be made more precise in Section 2.1 and 2.2:

o	The expectation operation E(.) is normally applied to random variables, not distribution functions \nu. I’d change the writing here – since X is not defined later anyway

o	Do players select a in an unrestricted way from A^M?

o	Is b common for all players?

o	The notation X is undefined

o	I think it would improve clarity to index b as b_t from its introduction in 2.1

o	You define the regret here, and then refer to bounds on the regret later, but presumably you bound the expected pseudo-regret rather than the random variable, the regret?

o	S_{a,b}(t) should be defined as (L_{a,b}(t),U_{a,b}(t)) with the lower limit of the interval first rather than the upper?

-	A legend on Figure 1, and some differentiation between the lines based on something other than colour would be beneficial for all readers (especially  colourblind).

-	It could be clearer whether the red curve is Algorithm 1 or not. If yes, better not to assume the reader is confident about the equivalence. If not, a comment would be useful to explain why it is not necessary to evaluate it.

---

> ### Author Response · Authors · 2026-03-05
>
> We sincerely thank the reviewer for their insightful feedback and detailed comments on our work. All major comments were fixed in the current uploaded manuscript, and we address each point below:
>
> **Major Comments:**
> 1. Thank you for raising this point. We have included a more detailed comparison of other works in our revised manuscript to better highlight the novelty of our work.
>
> **Proofs:**
> 1. Thank you for pointing this out; we've fixed this in the revision.
> 2. You are correct that the bound for the probability of the bad event should include a union across the players i; the corresponding theorem statements have been corrected to reflect this.
> 3. We have clarified the lemma statement with some details from the algorithm.
> 4. We have clarified the proofs accordingly.
> 5. Yes, showing that the regret is less than the sum of delta terms was an intermediate step in the proof, and the final complete regret bound does indeed include the 2BC term. Apologies for the confusion - we've revised the section slightly to be more organized.
> 6. From line 14 to 15, we use the fact that  $\\mu_\{\\star_\{c_t\}, c'\} - \\mu_\{\\star_\{c'\},c'\} \\leq 0,$ since $\\star_\{c'\}$ is the optimal arm for $c'$, which is why the middle term disappears in equation 15. The left and right terms are bounded in the paragraph above. From 15 to 16, there is a small change in notation from $\\star_\{c'\}$ to $a_t$ in the second term to emphasize the fact that the actions chosen at each round must be the optimal arm for some cluster (possibly not the real cluster, hence c'). We've revised the textual explanations in the proof to be more clear.
> 7. In Problem B, sabotage is triggered even under the good event, since as soon as any player notices that their confidence intervals for a given cluster have shrunk sufficiently to where they can eliminate that cluster from being admissible, they sabotage. This occurs even if not all players have realized that the cluster is no longer admissible. This allows players to learn from other players' rewards and speed up exploration, even without being able to observe others' rewards explicitly.  We can bound $N_s  \\leq BC,$ since players start by assuming each of the C clusters is admissible for each of the B nature's actions.
> 8. Yes, the bound is somewhat loose for our experiments, although this may be because our environment is relatively well-behaved. We will include more experiments with less well-behaved environments and a discussion of this discrepancy in the final version.
>
> **Minor:**
>
> 9. We will address these points in the final version. We appreciate the reviewer's comments on the presentation of our work.

---

> > ### Comment · Reviewer_2RHz · 2026-03-06
> > **Review of changes**
> >
> > Dear Authors,
> >
> > Thank you for your systematic reply and updates to the paper. Regarding literature review, and inconsistencies in the proofs (proofs points 1-6), I find the revisions made quite satisfactory and am pleased to consider my concerns handled in those areas. Regards to sabotage in Problem B, my confusion was how line 12 of algorithm 2 is what triggers a sabotage - i.e. where an arm mean falls outside a confidence interval, I had thought this was the same as the complement of the good event, but perhaps I am misunderstanding a notational subtlety somewhere?
> >
> > To sign off on the final part about the slackness of the bound - I would like to know some more details on the nature of these experiments, I guess they may be in progress at present, but some rough details would be helpful for having a think about it.
> >
> > Many thanks.

---

> > > ### Author Response · Authors · 2026-03-12
> > >
> > > We thank the reviewer for their positive comments on our revisions.
> > >
> > > Regarding problem B, the key point is that sabotage is triggered whenever $\\mu\_{a,c} \not\in S\_{a,b}(t),$ (where $S\_{a,b}(t)$ is the empirical confidence interval) whether or not b belongs to c. The complement of the good event is more specific: it occurs where a nature's action $b$ *does* belong to cluster $c$, but the empirical confidence interval $S\_{a,b}(t)$ does not contain the true mean $\\mu\_{a,c}$. So players may incorrectly interpret that as a signal that $b \not\in c$ and incorrectly eliminate $b$ from the set of admissible clusters for $c.$
> > >
> > > About slackness of the bound: we are currently running experiments with environments where variance is greater, the distribution of nature's actions is nonstationary across $T,$ and/or there are more clusters relative to the number of nature's actions. At present, the bound still appears somewhat slack. Currently we believe this is a consequence of the generality of our analysis, which assumes worst-case scenarios at several points to ensure the results hold across various instances. Tightening the bound would require introducing additional structural restrictions; we will add a discussion of this to better contextualize the empirical findings.

---

### Decision · Action_Editor_AfcW · 2026-04-08

**Recommendation:** Reject

**Additional Comments:**

The paper studies multiplayer latent bandits under asymmetric information and adversarial contexts, and the results are technically sound. However, two reviewers (hLQg and HbgZ) maintain leaning reject recommendations, citing limited novelty and strong assumptions. In particular, Reviewer HbgZ notes that the approach largely combines standard techniques, while Reviewer hLQg questions the distinction from existing formulations.

Another key concern is the looseness of the theoretical bounds. Reviewer 2RHz points out that the regret bounds appear quite slack and may not be meaningful at practical horizons, which weakens their interpretability and impact. While the authors clarified several issues in the rebuttal and improved presentation (acknowledged by Reviewer 2RHz), these responses do not substantially address the core concerns regarding novelty, assumptions, and the strength of the results.

Overall, I agree with the reviewers that the contribution is somewhat incremental and does not meet the bar for TMLR in its current form. I therefore recommend Reject.

**Audience:**

Yes

**Audience Explanation:**

All reviewers agreed that the topic is relevant and of potential interest to the TMLR audience. In particular, Reviewer 2RHz noted that the paper combines latent bandits and multiplayer settings in a meaningful way, while the other reviewers also acknowledged the general importance of the problem.

**Claims And Evidence:**

No

**Claims Explanation:**

While the paper provides formal analyses and proofs, the overall evidence is not fully convincing. In particular, Reviewer 2RHz points out that the regret bounds are quite loose and may not be meaningful at practical time horizons, raising concerns about their interpretability. Reviewer hLQg also questions the clarity of key aspects of the formulation (e.g., adversarial actions), and Reviewer HbgZ raises concerns about strong modeling assumptions that limit the applicability of the results. Although the authors clarified several points in the rebuttal, the combination of loose theoretical guarantees, limited empirical validation, and strong assumptions makes the overall evidence less compelling.

**Resubmission Of Major Revision:**

The authors may consider submitting a major revision at a later time.